# Evaluation of a human mucosal tissue explant model for SARS-CoV-2 replication

**Bhavna Gowan Gordhan[1]ʘ, Carolina Herrera[2]ʘ, Azure-Dee Pillay[3], Thabiso Seiphetlo[3], Christopher Shawn Ealand[1], Edith Machowski[1], Natasha Singh[3], Ntombexolo Seatholo[3], Kennedy Otwombe[3,4], Limakatso Lebina[3], Rebecca Frise[2], Gabriella Scarlatti[5], Francesca Chiodi[6], Neil Martinson[3,7‡], Julie Fox[8‡], Bavesh Davandra Kana🔟[1‡]\***

1 Faculty of Health Sciences, Department of Science and Innovation/National Research Foundation Centre of Excellence for Biomedical TB Research, School of Pathology, University of the Witwatersrand and the National Health Laboratory Service, Johannesburg, South Africa, 2 Department of Infectious Disease, Imperial College London, London, United Kingdom, 3 Faculty of Health Sciences, Perinatal HIV Research Unit (PHRU), University of the Witwatersrand, Johannesburg, South Africa, 4 Faculty of Health Sciences, School of Public Health, University of the Witwatersrand, Johannesburg, South Africa, 5 Viral Evolution and Transmission Unit, IRCCS Ospedale San Raffaele, Milan, Italy, 6 Department of Microbiology, Tumor and Cell Biology, Karolinska Institutet, Solna, Sweden, 7 Johns Hopkins University Center for TB Research, Baltimore, MD, United States of America, 8 Guys and St. Thomas' NHS Foundation Trust and King's College London, London, United Kingdom

ʘ These authors contributed equally to this work.
‡ These authors also contributed equally to this work.
* bavesh.kana@wits.ac.za

**Data Availability Statement:** All relevant data are within the paper and its Supporting Information files.

## Abstract

With the onset of COVID-19, the development of *ex vivo* laboratory models became an urgent priority to study host-pathogen interactions in response to the pandemic. In this study, we aimed to establish an *ex vivo* mucosal tissue explant challenge model for studying SARS-CoV-2 infection and replication. Nasal or oral tissue samples were collected from eligible participants and explants generated from the tissue were infected with various SARS-CoV-2 strains, including IC19 (lineage B.1.13), Beta (lineage B.1.351) and Delta (lineage B.1.617.2). A qRT-PCR assay used to measure viral replication in the tissue explants over a 15-day period, demonstrated no replication for any viral strains tested. Based on this, the *ex vivo* challenge protocol was modified by reducing the viral infection time and duration of sampling. Despite these changes, viral infectivity of the nasal and oral mucosa was not improved. Since 67% of the enrolled participants were already vaccinated against SARS-CoV-2, it is possible that neutralizing antibodies in explant tissue may have prevented the establishment of infection. However, we were unable to optimize plaque assays aimed at titrating the virus in supernatants from both infected and uninfected tissue, due to limited volume of culture supernatant available at the various collection time points. Currently, the reasons for the inability of these mucosal tissue samples to support replication of SARS-CoV-2 *ex vivo* remains unclear and requires further investigation.

**Funding:** The primary study was supported by funding from the European and Developing Countries Clinical Trials Partnership (EDCTP), grant number RIA2020EF-3008 (to JF). The laboratory analysis of the specimens was supported by funding from the South African National Research Foundation DCOE-015 (to BDK), the South African Medical Research Council with funds from the Department of Health (to BDK) and the National Health Laboratory Service Research Trust (to BGG). The funders had no role in the study design, data collection and analysis, decision to publish, or preparation of the manuscript.

**Competing interests:** The authors have declared that no competing interests exist.

## Introduction

As pandemics such as COVID-19 emerge, it is imperative to rapidly develop physiologically relevant laboratory models that allow for evaluation of pathogen infection, host-pathogen interactions, immune modulation, and evaluation of potential novel therapeutics. Conventionally, cell lines and animal models have been the mainstay in understanding molecular mechanisms of pathogenesis for infectious agents [1–3]. However, there are inherent limitations to these approaches including an inability to recapitulate the complexities and salient features of human infection, high costs, and reliance on specialized laboratories with highly trained personnel. More recently, human *ex vivo* challenge models using tissue from different anatomic sites together with the development of three-dimensional (3D) organ cultures from stem cells (organoids) or organ-on-a-chip technology, have revolutionised the study of disease physiopathology [4–6]. These technologies offer a more realistic avenue for cheaper and accelerated testing of potential therapeutics and triage of vaccine candidates prior to preclinical and clinical evaluation [5,7,8]. *Ex vivo* cultures of human lung tissue have been used extensively to investigate the replication, cell tropism, and immune activation profile of novel respiratory pathogens [9]. For example, lung tissue infected with SARS-CoV or SARS-CoV-2 strains demonstrated that the latter replicated more efficiently and produced three times more viral particles [10]. These findings might explain the high viral loads in the respiratory secretions of COVID-19 patients during early infection and thus the high transmissibility. SARS-CoV-2 elicited a reduced pro-inflammatory response relative to SARS-COV, accounting for the mild or lack of symptoms in many COVID-19 patients who as a result unknowingly continue to spread the virus, making it difficult to contain the pandemic [10]. Similarly, both the Wuhan and the Beta strains showed comparable replication competency in explant tissues of the human upper respiratory tract and primary nasal epithelial cells with the variant inducing a reduced type III IFN response in nasal cells [11]. A cryopreserved bank of human lung tissue from 21 donors with clinical risk factors for severe COVID-19 also allowed for delineating inter-individual heterogeneity of the host response to SARS-CoV-2. Cryopreserved tissues containing heterogenous populations of metabolically active epithelial, endothelial, and immune cell subsets of the human lung were susceptible to SARS-CoV-2 infection which revealed a donor-dependent heterogeneity in the expression of select immune markers in response to SARS-CoV-2 [12]. In another study, nasal mucosa and lung tissues infected with SARS-CoV-2, showed active viral replication and differential tissue specific innate immune responses in the upper and lower respiratory tract [13]. Studies such as these provide important insights on the pathogenesis, transmissibility, and asymptomatic infection of respiratory viruses.

To understand whether immune evasion or intrinsic virological properties were responsible for the exceptional transmissibility of the SARS-CoV-2 Omicron variant, the replication competence and cellular tropism of the Wuhan, Beta, Delta and Omicron variants have been compared in explant cultures of human bronchi and lungs [14,15]. Omicron replicated faster than all other SARS-CoV-2 strains in tissue derived from the bronchi but less efficiently in the lung parenchyma tissue, possibly explaining the reduced severity of Omicron reported in epidemiological studies. In contrast, our prior work demonstrated that replication competency was equivalent for the Wuhan strain and several viral variants when assessed in Vero E6 cells [16]. These discordant observations confirm the value of *ex vivo* human tissue explant models in providing important biological correlates of viral pathogenesis. In the present study, we assessed whether SARS-CoV-2 strains were capable of infecting and replicating in nasal or oral tissue obtained from vaccinated and non-vaccinated individuals recruited in Johannesburg, South Africa [17].

## Material and methods

This study was approved by the University of Witwatersrand, Human Research Ethics Committee (Reference 200711). All methods were performed in accordance with the relevant guidelines and regulations for growing and handling of SARS-CoV-2 as approved by the Institutional BioSafety Committee of the University of the Witwatersrand (approval number: 20200502Lab). All experiments were conducted in a BioSafety Level III (BSL3) laboratory, registered with the South African Department of Agriculture Forestry and Fisheries (registration number: 39.2/NHLS-20/010) with the use of appropriate personal protection equipment to prevent transmission of infection to personnel.

### Recruitment of participants to obtain oral or nasal tissue specimens

A cross-sectional study was conducted recruiting patients from June 2021 through June 2022, at the Donald Gordon Hospital (Parktown) in South Africa to recruit adults 18–70 years old with a recent SARS-CoV-2 negative test. Individuals undergoing elective maxillofacial surgery for anatomical correction were approached for written consent to donate residual nasal or oral tissue for analysis. All personal information and data linked to the participants was anonymized.

A total of 18 participants with a negative SARS-CoV-2 test prior to surgery were recruited over the one-year study period immediately prior to their surgical procedure. Time of resections was accurately recorded and residual surgically resected oral and nasal tissues were immediately transferred into 10 ml of cold complete medium DMEM/F-12 (Dulbecco's Modified Eagle's Medium supplemented with 2 mM L-glutamine and antibiotics (100 U of penicillin/mL, 100 μg of streptomycin/mL, 0.25 μg of streptomycin/mL, and 80 μg of gentamicin/mL)) and transported on ice to be received by the laboratory within an hour of resection. Once received at the laboratory the explant tissue was prepared immediately for *in vitro* infection challenge. Tissue from the first participant was used to establish the laboratory protocol and was mock infected with media. Explants generated from tissue obtained from the remaining 17 participants were challenged with SARS-CoV-2 using different variant strains and protocol modifications as described later.

### SARS-CoV-2 viral strains and culture conditions

Purified SARS-CoV-2 strains including IC19 (lineage B.1.13), Beta (lineage B.1.351) and Delta (lineage B.1.617.2), were received from Imperial College, London as dry ice shipments. Upon arrival, the viability and integrity of the IC19 strain was confirmed by infecting Vero E cells and quantifying nucleic acid material in the supernatant, 4 days after challenge by qRT-PCR as described previously [16]. Briefly, one vial of the virus was thawed and diluted 10-, 100- or 1000-fold to represent high, medium, and low viral loads, respectively. Vero E6 cells seeded at $1.2 \times 10^6$ in a 24-well microtitre plate were infected with 250 μl of the respective viral dilution and incubated at 37°C for one hour. After infection, 250 μl of complete DMEM was added to each well and the microtitre plate incubated in a $CO_2$ incubator for 4 days. Immediately after adding the media, 150 μl of the supernatant was removed, and heat inactivated for 5 minutes at 70°C in 600 μl of lysis buffer for qRT-PCR analysis. The harvested supernatant was replaced with fresh complete DMEM. After 4 days of incubation, approximately 80% of the cells showed detachment. At this time, 150 μl of the supernatant was removed for RNA extraction and quantification of viral copy number by qRT-PCR.

### Infection of explant tissue with SARS-CoV-2

Upon receipt in the lab, the resected tissue was placed in a Petri dish containing 1 ml of complete DMEM/F-12. The tissue was immediately dissected into multiple 2–3 mm³ explants and

two explants were transferred into a single well, in a 96-well plate containing 100 μl of complete DMEM/F-12 in each well. Explants were challenged with 100 μl of virus at various titres (high ($10^5$ pfu/ml), medium ($10^4$ pfu/ml), low ($10^3$ pfu/ml)) of the respective SARS-CoV-2 strains, either in duplicate or triplicate depending on the availability of explants. An uninfected well containing only the explant in 200 μl of complete DMEM/F-12 was included in each experiment to serve as a negative control. The plate was incubated for 20 h at 37˚C in a $CO_2$ incubator. The following day, 110 μl of culture supernatant (pre-wash) was removed from each well and frozen for further analysis. Each explant was then washed to remove unbound virus by sequentially transferring the explants with a sterile forceps from the incubation well to 4 wash wells containing 200 μl of PBS. The washed explants were placed in a new 96-well plate containing 200 μl of complete DMEM/F-12 in each well. A 110 μl aliquot of culture supernatant from the challenge (post-wash supernatant) and negative control wells was collected, and the respective wells were replenished with 110 μl of fresh complete DMEM/F-12. The explants were incubated for 24 h at 37˚C in a $CO_2$ incubator before placing them in a 24-well plate containing gelatine rafts saturated in complete DMEM/F-12. Plates were incubated for a further 15 days at 37˚C with 5% $CO_2$. Aliquots of the culture supernatant (110 μl) were harvested at various timepoints (days 1, 2, 3, 7, 11 and 15) post infection for nucleic acid quantification using qRT-PCR. At each harvest time, the volume of supernatant removed from each well was replaced with freshly prepared complete medium.

## Modified protocol for infection of explant tissue with SARS-CoV-2

The preparation of the explant from the nasal or oral tissue was as described above. However, challenge time with the virus was reduced to 2 hrs, with experiments done either in duplicate or triplicate depending on the availability of explants. A control sample, corresponding to an uninfected well containing only the explant in 200 μl of complete DMEM/F-12, was included in each experiment. After the 2 hr incubation period, 200 μl of culture supernatant (pre-wash) was removed for qRT-PCR. The infected explants were then washed 2 times in a 48-well microtitre plate containing 1 ml of PBS to remove unbound virus. Post washing, the explants were placed directly in 300 μl DMEM/F-12 media without gelatine rafts and 250 μl of culture supernatant was collected for qRT-PCR from the negative controls and challenge wells (post-wash supernatant). The respective wells were replenished with 250 μl of freshly prepared complete DMEM/F-12. Thereafter, 140 μl aliquots of the culture supernatant were harvested at days 1 and 2 post infection for qRT-PCR analysis. In this case, the volumes, corresponding to aliquots taken, were not replaced with complete medium.

## Extraction of total RNA and cDNA synthesis

All RNA extractions were performed using the NucleoSpin Viral RNA kit (Macherey-Nagel). Briefly, the supernatants (90–150 μl) containing viral particles were mixed with 600 μl RAV1 lysis buffer (45–60% guanidium thiocyanate) containing carrier RNA and heat inactivated at 70˚C for 5 min in the BSL3 laboratory. RNA was then extracted as per the manufacturer's instruction. To ensure that heating did not affect viral quantification, the RNA extraction procedure was carried out with and without the heat inactivation step in the BSL3 laboratory. For the SYBR green reactions, 23 μl of total RNA (out of 50 μl) was mixed with 2 μl of a reverse primer mix (E gene-specific primers at a final concentration of 2.5 μM) to generate complimentary DNA (cDNA). Primers were annealed to the RNA (94˚C for 1.5 min, 65˚C for 3 min, 57˚C for 3 min) and then snap-cooled on ice. A two-step protocol was used to generate cDNA for testing in the Beta or Delta specific qPCR assay. Briefly, 5 μl of total RNA was mixed with 2 μl of a reverse primer mix (containing spike-specific primers at a final concentration of

2.5 μM) (S1 Table) made up to a final volume of 25 μl with sterile water. Primers were annealed to RNA (94°C for 1.5 min, 65°C for 3 min, 57°C for 3 min) and then snap-cooled on ice. The primer-annealed RNA sample (12.5 μl) was then added to a 12.5 μl reaction containing Super-script IV (reverse transcriptase (RT), Invitrogen). cDNA was synthesized by incubating reactions at 55°C for 15 min followed by heat-inactivation at 80°C (as per the manufacturer's instructions).

## qPCR analysis to test infectivity of SARS-CoV-2 for explant infection

To assess infectivity of viral stocks prior to explant challenge, a SYBR green (Agilent, Diagnostech) based qPCR analysis was carried out on a Bio-Rad CFX96 real-time PCR machine (C1000 touch thermal cycler) as previously described [16]. Standard curves were generated for the $E$-gene primer set as described previously [18] (S1 Table) by creating a dilution series ($10^7$ to $10^1$ copies/reaction) of known concentrations of plasmid DNA (BN2) harboring the $E$-gene. One microliter of each sample (BN2 standards, cDNA sample and a no template control [NTC]) was assessed in 20 μl volumes using an optimized thermal cycling profile (98°C for 2 min and 40 cycles of 98°C for 5s, 59°C for 5s, 72°C for 5s). Following qPCR, amplification specificity was determined by a melt curve analysis. Each reaction was performed in duplicate. Having established that the minor groove binder (MGB) probes were specific for either the Wuhan or Beta variant, we only used the Beta-variant MGB probe for subsequent testing on culture supernatant samples. The Brilliant III Ultra-Fast QPCR Master Mixes (Agilent) were used in all assays. Briefly, 1 μl of each cDNA sample was assessed in 20 μl volumes using an optimized thermal cycling profile (95°C for 3 min; 40x cycles (95°C for 15s, 60°C for 60s plus a plate reading in the appropriate channels (Fam, Hex, Cy5 or Rox)). A standard curve corresponding to $10^7$ and $10^0$ genome equivalents enabled transcript quantification. In addition, a NTC and positive control (cDNA derived from a confirmed clinical Beta strain) [16] was used in every run. Each reaction was performed in duplicate, and all qPCR assays were performed on a Bio-Rad CFX96 real-time PCR machine (C1000 touch thermal cycler). Data analysis and the corresponding graphs were generated using Microsoft Excel, version16.71.

## Design of qRT-PCR assay to distinguish the E484K or L452R mutations in the spike gene

The E484K mutation (GAA → AAA) present in the spike gene of the Beta-variant (also known as 20H and B.1.351) was targeted for qRT-PCR assay development (S1A Fig). The P681R mutation (CCT → CGT) present in the spike gene of the Delta-variant (also known as 21A and B.1.617.2) was similarly targeted for assay development (S2A Fig). DNA sequences corresponding to the flanking primers and probes, together with the optimized PCR cycling conditions are listed in S1 Table. To determine primer specificity, reactions containing plasmid DNA corresponding to the Beta or Delta-variants and primers flanking the SNP were performed using the following parameters: 98°C for 2min; 40x cycles (98°C for 5s, 59°C for 5s, 72°C for 5s plus plate read); melt curve at 65–98°C with 0.5°C increment and plate read between each step. Single peaks in the melt curve analyses were indicative of adequate primer specificity (S1B and S2B Figs). Following confirmation of primer specificity, MGB probes were included in a multi-plex qPCR assay. MGB probes bind in a hydrophobic way to the DNA helix which increases the $T_M$ by 15–30°C enabling the use of shorter probes with increased specificity [19,20]. Specificity of both MGB probes were confirmed using cognate sequences corresponding to either IC19, Beta- or Delta-variant (S1C and S2C Figs, respectively). The supernatants harvested from the explant infections with SARS-CoV-2 strains were tested by

qRT-PCR using these optimized MGB probes. qRT-PCR data analysis and the corresponding graphs were generated using Microsoft Excel, version16.71

## Results

### Infectivity of SARS-CoV-2 virus used for explant challenge

Viral stocks of the IC19 strain were diluted 10-fold (high titre), 100-fold (medium titre) or 1000-fold (low titre) and an aliquot of each was used to infect Vero E cells. RNA extracted from an aliquot of the neat and diluted virus was analyzed by qRT-PCR to evaluate the integrity of the virus. The neat virus showed titers of approximately $10^6$ RNA molecules/reaction (rxn) with 10-fold reduction in RNA, with increased dilution of the virus (S3 Fig–panel A). The replication competency of the transported virus was verified by infecting Vero E cells with an aliquot of the neat and diluted virus. The virus retained infectivity after 96 hrs of challenge, as the number of RNA molecules detected increased by ca. 3 log ($10^6$ RNA molecules) relative to the inoculum ($10^3$ RNA copies) at the time of challenge (0 hrs) (S3 Fig—panels B and C).

### *Ex vivo* challenge of nasal and oral tissue explants

As we were unsure of the viral titre in the culture supernatants from the infected tissue explants, we had to ensure that the RNA extraction method did not compromise the yield and integrity of the RNA. We hypothesized that the heat inactivation in the extraction protocol could affect RNA yields. Hence, the neat SARS-CoV-2 virus was diluted 10-fold from $10^6$–$10^1$ and the RNA extracted with and without heat inactivation. The amount of viral RNA recovered by the two methods was compared by qRT-PCR, which demonstrated no significant difference under both conditions of extraction (S4 Fig). All downstream RNA extractions therefore, included the heat inactivation step as it provided an additional layer of safety during the extraction process.

We aimed to establish a mucosal tissue explant model of *ex vivo* challenge to investigate infection by SARS-CoV-2 across the oral and nasal mucosa. Our target cohort was individuals unvaccinated against SARS-CoV-2 however, due to ethical clearance delays, 67% of the participants enrolled had received the SARS-CoV-2 vaccination by the time the study started (Table 1). To circumvent this limitation, we opened more recruitment centres in the region but this did not improve the recruitment of unvaccinated individuals. A total of 18 adult participants aged between 18 and 70 years with a negative SARS-CoV-2 test prior to surgery were recruited to provide residual nasal or oral tissue. The baseline characteristics for the participants are described in Table 1 and in our recent study [17]. Briefly, the cohort was 61% male and 39% female with a median age of 40.2 years and an average BMI of 27.3. None of the participants were infected with HIV and 67% were vaccinated (11/12 with BioNTech/Pfizer and 1/12 with J&J) against SARS-CoV-2. The average time from vaccination to recruitment into the study was 19.8 (±14.9) weeks.

Oral and/or nasal tissue explants from the first ten participants were infected with 3 different dilutions of SARS-CoV-2 (IC19 strain) for 20 hrs and post washing, the explants were placed on gelatin rafts to facilitate replication of the virus. Aliquots of the culture supernatant were harvested at various time points up to 15 days and stored at -80˚C for downstream processing as shown in Fig 1A. To assess viral replication, RNA from the stored culture supernatants was purified and analysed by qRT-PCR. During the sampling period, we considered a viral titre that is significantly higher than that detected at day 0 or day 1 as evidence of viral replication. We set the limit of accurate detection at 50 viral copies/rxn. Explants obtained from the first 10 individual participants infected with high, medium or low titre of virus showed no replication of the virus in the mucosal tisssue over the 15 days as the RNA

**Table 1. Baseline characteristics of COVAB study participants.**

| Characteristics | Participants (n = 18) |
|---|---|
| Gender (%) | Female 7 (39)<br>Male 11 (61) |
| Age (years), mean (SD) | 40.2 (14.8) |
| BMI, mean (SD) | 27.3 (5.6) |
| HIV Status (%) | Negative 18 (100)<br>Positive 0 (0) |
| COVID-19 symptoms (cough, fever, shortness of breath, vomiting diarrhoea, headache, loss of taste and smell) (%) | No 18 (100)<br>Yes 0 (0) |
| COVID-19 vaccinated (%) | No 6 (33)<br>Yes 12 (67) |
| Type of COVID-19 vaccine (%) | J&J 1 (8)<br>BioNTech/Pfizer 11 (92) |
| Time since vaccination (weeks), mean (SD) | 19.8 (14.9) |
| Previous positive COVID-19 test (%)* | No 14 (78)<br>Yes 4 (22) |
| Vaccinated and infected (%)§ | No 8 (67)<br>Yes 4 (33) |
| Currently smoking (%) | No 13 (72)<br>Yes 5 (28) |
| Diabetes (%) | No 18 (100)<br>Yes 0 (0) |
| Hypertension (%) | No 15 (83)<br>Yes 3 (17) |
| Asthma and/or chronic obstructive pulmonary disease COPD (%) | No 12 (67)<br>Yes 6 (33) |
| Smoker (%) | No 13 (72)<br>Yes 5 (28) |

*Positive test at least 4 months before surgery.

§Vaccinated individuals testing positive for COVID-19 at least 4 months before surgery. % calculated for number of vaccinated participants.

transcripts were equal to or below the transcripts detected at day 1 or were below the limit of detection (values were <50 copies/rxn; S5–S8 Figs). As expected, approximately 1000–10 000 copies of RNA molecules were detected in the pre-wash culture media. However, once the explants were washed and placed in freshly prepared media, the number of detectable viral RNA copies diminished to approximately 50–100 RNA copies. Only infected tissue explants from particpant 5 (S6 Fig) showed the presence of RNA transcripts at days 7, 11 and 15. However, the transcript levels detected at these timepoints were not significantly higher than that detected at day 1 (p = 0,7; 0,2 and 0,1; Mann-Whitney t-test comparing day 1 with days 7, 11 and 15 respectively) suggesting that the explant did not support viral replication over time. The average viral titres from the explants of the 10 participants is shown in Fig 1B.

As the IC19 strain showed no infectivity of nasal or oral tissue, we next tested whether the Beta and Delta variant strains had the potential to infect mucosal tissue. Oral tissue from two participants was infected with either the IC19, Beta or Delta strains as per the protocol described in S9A Fig. Supernatants from the infected explants were harvested at the various time points up to 15 days as before. Based on previous observations that showed no viral replication after 1–3 days, here we initially analysed samples collected up to 3 days by qRT-PCR. Infected tissue explants from the two participants did not support viral replication of both the

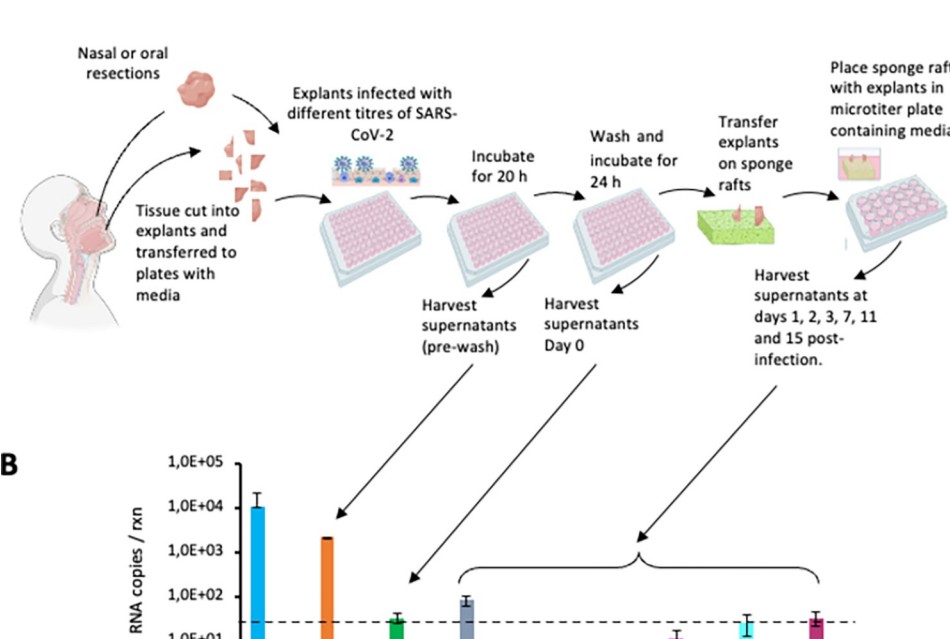

**Fig 1. *Ex vivo* challenge of oral and nasal tissue with SARS-CoV-2.** (A) Experimental protocol used to infect oral or nasal tissue with SARS-CoV-2. The tissue was infected with the IC19 strain for 24 hrs after which, it was washed and incubated for a further 24 hrs before transferring the tissue onto sponge rafts. At various time points supernatants were harvested for downstream analysis. (B) Assessment of the culture supernatant for viral replication of SARS-CoV-2 (IC19 strain) in oral or nasal tissue over a 15 day period from 10 participants by qRT-PCR. The data represent the average from the 10 participants. The error bars represent the standard deviation between experiments. The dotted line represents the limit of detection (50 copies / reaction). NV = neat virus, Control = uninfected explants, High = high titre virus, 20 hrs = sample called pre-wash of explants. Fig 1A was generated using Biorender (https://www.biorender.com/) and is for illustrative purposes only.

Beta and Delta variants after 3 days (S9 and S10 Figs). Hence, samples taken at later time points were not analysed further by qRT-PCR.

## *Ex vivo* challenge of nasal and oral tissue explants using the modified protocol

As the *ex vivo* infection of nasal and oral explants proved to be unsuccessful, we modified the protocol as follows: 1) the viral challenge period was shortened to 2 hrs; 2) the gelatine rafts were removed and the explants were incubated directly in the culture media; 3) it was also possible that 2 explants may have limited targets for infection or may produce limited amount of *de novo* virus hence, each infection was carried out simultaneously with 2 and 4 explants in the same volume of complete media, and 4) the harvest period of culture supernatants was shortened to 24 hrs and 48 hrs post challenge for qRT-PCR analysis. The revised protocol is shown in Fig 2A. Despite these modifications, no productive infection was observed in tissue explants from five participants (S11 and S12 Figs). The average RNA transcripts for the 5 participants is shown in Fig 2B.

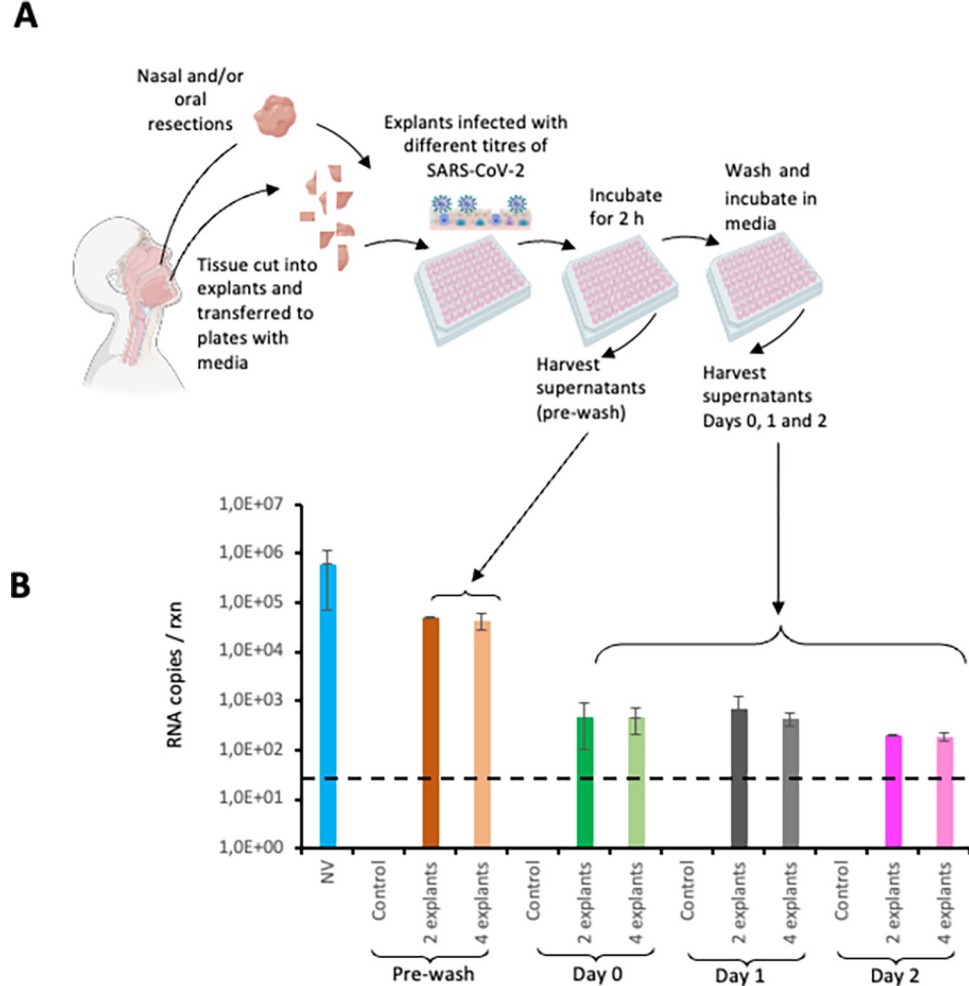

**Fig 2.** *Ex vivo* **challenge of oral and nasal tissue with SARS-CoV-2.** (A) Experimental protocol 2 used to infect oral or nasal tissue with SARS-CoV-2. The tissue was infected with the virus for 2 hrs after which it was washed and incubated for 48 hrs in liquid media. Samples were collected at 24 hrs and 48 hrs. (B) Assessment of the culture supernatant for viral replication by qRT-PCR. The data represent the average of 5 participants. Each experiment was set up in triplicate where possible depending on the size of the tissue received. The error bars represent the standard deviation between experiments. The dotted line represents the limit of detection (50 copies / reaction). NV = neat virus, Control = uninfected explants. Fig 1A was generated using Biorender (https://www.biorender.com/) and is for illustrative purposes only.

## Discussion

The upper respiratory tract is the primary entry site for respiratory viruses, including SARS-CoV-2 however, infectious pathways and pathogenesis at these mucosal surfaces is not fully understood. Recently, human nasal turbinate tissue and nasal epithelial cultures have been used to identify tissue-specific and virus-specific aspects of early infection of SARS-CoV-2, with parallel investigations of the virus in lung tissue, the major end-organ site for viral replication and disease [11,13]. These studies have demonstrated active replication of SARS-CoV-2 with comparable infection kinetics in both the nasal and lung tissue. Our methodology for tissue maintenance and infection was similar to that described in these studies that used Roswell Park Memorial Institute 1640 medium (RPMI) for nasal tissue and DMEM-F12 for lung tissue. We chose to use DMEM-12 since testing of four culture media on explant attachment, growth,

and phenotype of oral mucosal epithelial cells showed that the choice of media had no impact on the ability of the explants to attach to the culture well with DMEM-F12 and RPMI, with both supporting the largest cellular outgrowth [21]. Cryopreserved human lung tissue used to investigate pathogenesis and drug screening also showed 90% cell viability and demonstrated replication of SARS-CoV-2 [12]. In contrast, our study wherein we attempted to establish an *ex vivo* challenge model for SARS-CoV-2 using either nasal or oral mucosal tissue did not yield viral replication, despite repeated methodological changes to align our protocol to metodologies reported in the litterature. Our initial protocol used gelatine sponge rafts to suspend the tissue explant and the replication of the SARS-CoV-2 was measured over a 15 day period. Under these conditions infections in both the oral and nasal tissue were unsuccessful as the number of RNA copies in the culture supernatants of the infected explants did not increase over the 15 day period for all three strains of the SARS-CoV-2 virus (IC19, Beta and Delta). Based on these data we hypothesized that suspension of the explants on the gelatine rafts may have led to tissue necrosis resulting in cell death with a resultant loss of the necessary receptors required for attachment and successive rounds of viral replication. We therefore, modified the protocol to remove the gelatine sponges and shorten the exposure time of the tissue and virus by 10 fold. These modifications still did not improve the infectivity and replication competency of SARS-CoV-2.

Due to delays in receiving ethical approval, recruiting unvaccinated participants was difficult and as such, two thirds of the consenting participants were vaccinated. It would be interesting to test whether circulating neutralizing antibodies and/or other host molecules/factors present in the tissue explant maybe neutralizing the virus, thus preventing viral replication. Side by side comparison of samples collected from *ex vivo* challenge tissue from both vaccinated and unvaccinated individuals will improve our understanding of viral replication or lack thereof in *ex vivo* challenge models. However, due to the limited availability of tissue and culture supernatant from participants enrolled in the study, we were unfortunately unable to investigate this apect further. Although at present, the reason for the lack of SARS-CoV-2 strains to infect oral or nasal tissue explants is unclear and requires further optimization, our study does provide valuable iterative methodological approaches that can be used to improve future studies aimed at using human explants.

## Supporting information

**S1 Fig. Optimisation of E484K multiplex qPCR assays for Beta variant calling.** (A) Schematic of genome organization of SARS-CoV-2 coronavirus showing primer pairs and MGB probes targeting the E484K SNP within the Spike gene. Enlarged region corresponds to the E484K SNP (blue line within brown box) with flanking primers (S-F1 and S-R2 depicted by green boxes) and probes (brown box) for multiplex assay (cognate sequence for Wuhan-specific probe = 484-Wuhan fluorescently labelled with Cy5; cognate sequence for Beta-specific probe = 484-Beta fluorescently labelled with Rox). (B) Amplification plot and melt curve analysis (SYBR green chemistry) of S-F1R1 primer pair to confirm primer specificity using cDNA derived from Wuhan-Hu-1. A single peak in the melt-curve analysis is indicative of primer specificity. (C) Amplification curves of fluorescence intensity versus cycle threshold (CT) with E484K primer-probe sets of multiplex qPCR. The Wuhan-Hu-1 (Cy5) and Beta (Rox) MGB probes targeting the variant-defining SNPs were combined in a single reaction and tested using cDNA derived from either of the two strains. Both MGB probes were highly specific and only amplified cDNA containing its cognate SNP. The no template control reactions showed no amplification and were void of any contaminating DNA. All qPCR reactions were set up in duplicate. (TIF)

**S2 Fig. Optimization of P681R multiplex qPCR assays for Delta variant calling.** (A) Schematic of genome organization of SARS-CoV-2 coronavirus showing primer pairs and MGB probes targeting the P681R SNP within the Spike gene. Enlarged region corresponds to the P681R SNP (blue line within brown box) with flanking primers (681-F3 and 681-R3 depicted by grey boxes) and probes (brown box) for multiplex assay (cognate sequence for Wuhan-specific probe = 684-Wuhan fluorescently labelled with Rox; cognate sequence for Delta-specific probe = 681-Delta fluorescently labelled with Cy5). (B) Amplification plot and melt curve analysis (SYBR green chemistry) of 681_F3R3 primer pair to confirm primer specificity using cDNA derived from the Delta strain. A single peak in the melt-curve analysis is indicative of primer specificity. (C) Amplification curves of fluorescence intensity versus cycle threshold (CT) with P681R primer-probe sets of multiplex qPCR. The Wuhan-Hu-1 (Rox) and Delta (Cy5) MGB probes targeting the variant-defining SNPs were combined in a single reaction and tested using cDNA derived from either of the two strains. Both MGB probes were highly specific and only amplified cDNA containing its cognate SNP. The no template control reactions showed no amplification and were void of any contaminating DNA. All qPCR reactions were set up in duplicate.
(TIF)

**S3 Fig. Assessment of SARS CoV-2 infectivity.** (A) The frozen sample of SARS-CoV2 received from ICL (IC19) was thawed, diluted 10-fold, 100-fold and 1000-fold, and assessed by qRT-PCR for genome integrity. Vero E cells were infected with the three dilutions of SARS-CoV-2 (from A) to assess the viability and infectivity of the virus. (B) Culture supernatants were assessed immediately after infection by qRT-PCR, T = 0 and (C) 96 hrs after infection. The error bars represent the standard deviation of two replicates.
(TIF)

**S4 Fig. Assessment of SARS CoV-2 RNA integrity.** The neat virus was diluted ($10^6$–101—represented as 1–6 on the x-axis of the graph) and RNA from 2 aliquots of each dilution was extracted as per the manufacturer's recommendation except the one aliquot of the virus was not heat inactivated at 75°C for 5 min The number of RNA molecules extracted with and without heat treatment were compared by qRT-PCR. The error bars represents the standard deviation of two replicates.
(TIF)

**S5 Fig. *Ex vivo* challenge of oral and nasal tissue from study participants 2, 3 and 4 with SARS-CoV-2.** Explants of oral or nasal tissue from participants 2, 3 and 4 were infected with SARS-CoV-2, strain IC19 as per the protocol described in Fig 1A. After 24 hrs of infection the tissue was washed and incubated for a further 24 hrs before transferring the tissue onto sponge rafts. At various time points supernatants were harvested over a 15-day period for assessment of viral replication of SARS-CoV-2 by qRT-PCR. Each experiment was set up in triplicate where possible depending on the size of the tissue received. The error bars represent the standard deviation between experiments. The dotted line represents the limit of detection (50 copies / reaction). NV = neat virus, Control = uninfected explants, High = high titre virus, Medium = 10-fold dilution of the high titre virus, Low = 10-fold dilution of the medium titre virus and 20 hrs = sample collected pre-wash of explants.
(TIF)

**S6 Fig. *Ex vivo* challenge of oral and nasal tissue from study participants 5, 6 and 7 with SARS-CoV-2.** Explants of oral or nasal tissue from participants 5, 6 and 7 were infected with SARS-CoV-2, strain IC19 as per the protocol described in Fig 1A. After 24 hrs of infection the tissue was washed and incubated for a further 24 hrs before transferring the tissue onto sponge

rafts. At various time points supernatants were harvested over a 15-day period for assessment of viral replication of SARS-CoV-2 by qRT-PCR. Each experiment was set up in triplicate where possible depending on the size of the tissue received. The error bars represent the standard deviation between experiments. The dotted line represents the limit of detection (50 copies / reaction). NV = neat virus, Control = uninfected explants, High = high titre virus, Medium = 10-fold dilution of the high titre virus, Low = 10-fold dilution of the medium titre virus and 20 hrs = sample collected pre-wash of explants. For participant 5, a Mann Whitney t-test was used to determine significance between RNA transcripts detected at days 1–15, the resulting p-values are shown on top of each timepoint, ns = not significant.
(TIF)

**S7 Fig.** *Ex vivo* **challenge of oral and nasal tissue from study participants 8, 9 and 10 with SARS-CoV-2.** Explants of oral or nasal tissue from participants 8, 9 and 10 were infected with SARS-CoV-2, strain IC19 as per the protocol described in Fig 1A. After 24 hrs of infection the tissue was washed and incubated for a further 24 hrs before transferring the tissue onto sponge rafts. At various time points supernatants were harvested over a 15-day period for assessment of viral replication of SARS-CoV-2 by qRT-PCR. Each experiment was set up in triplicate where possible depending on the size of the tissue received. The error bars represent the standard deviation between experiments. The dotted line represents the limit of detection (50 copies / reaction). NV = neat virus, Control = uninfected explants, High = high titre virus, Medium = 10-fold dilution of the high titre virus, Low = 10-fold dilution of the medium titre virus and 20 hrs = sample collected pre-wash of explants.
(TIF)

**S8 Fig.** *Ex vivo* **challenge of oral and nasal tissue from study participants 11 with SARS-CoV-2.** Explants of oral or nasal tissue from participant 11 was infected with SARS-CoV-2, strain IC19 as per the protocol described in Fig 1A. After 24 hrs of infection the tissue was washed and incubated for a further 24 hrs before transferring the tissue onto sponge rafts. At various time points supernatants were harvested over a 15-day period for assessment of viral replication of SARS-CoV-2 by qRT-PCR. The experiment was set up in triplicate where possible depending on the size of the tissue received. The error bars represent the standard deviation between experiments. The dotted line represents the limit of detection (50 copies / reaction). NV = neat virus, Control = uninfected explants, High = high titre virus, Medium = 10-fold dilution of the high titre virus, Low = 10-fold dilution of the medium titre virus and 20 hrs = sample collected pre-wash of explants.
(TIF)

**S9 Fig.** *Ex vivo* **challenge of oral and nasal tissue with IC19, Beta and Delta SARS-CoV-2 strains.** (A) Experimental protocol used to infect oral or nasal tissue with the different SARS-CoV-2 strains. The explants were infected with either the IC19, Beta or Delta strain of SARS-CoV-2 for 20 hrs after which, the explants were washed and incubated for a further 24 hrs before transferring them onto sponge rafts. At various time points supernatants were harvested for downstream analysis. (B) Assessment of culture supernatant by qRT-PCR for viral replication of SARS-CoV-2 IC19, Beta and Delta strains over a 3 day period in oral tissue obtained from two participants. For each viral titre the experiment was set up in triplicate where possible depending on the size of the tissue received. The standard deviation is represented by ±. *ND = Not detected and refers to a viral load that is not significantly higher than that detected at day 0, or no virus detected at all. The samples collected at days 7, 11 and 15 were not analyzed by qRT-PCR.
(TIF)

**S10 Fig.** *Ex vivo* **challenge of oral and nasal tissue from study participants 12 and 13 with Wuhan, Beta and Delta SARS-CoV-2 strains.** Explants from participants 12 and 13 were infected with different titres of either the IC19, Beta or Delta strain of SARS-CoV-2 as per the protocol described in Fig 1A. For each viral titre the experiment was set up in triplicate where possible. Supernatants were harvested at various time points for assessment by qRT-PCR for viral replication. Each experiment was set up in triplicate where possible depending on the size of the tissue received. The error bars represent the standard deviation between experiments. The dotted line represents the limit of detection (50 copies / reaction). NV = neat virus, Control = uninfected explants, High = high titre virus, Medium = 10-fold dilution of the high titre virus, and 20 hrs = sample collected pre-wash of explants. The solid bars represent IC19, the stripe bars represent the Beta strain and the dotted bars represent the Delta strain. The samples collected at days 7, 11 and 15 were not analyzed by qRT-PCR.
(TIF)

**S11 Fig.** *Ex vivo* **challenge of oral and nasal tissue from study participants 14, 15 and 16 with SARS-CoV-2.** Two or four oral or nasal tissue explants from participants 14, 15 and 16 were infected with SARS-CoV-2, strain IC19 using the modified protocol as described in Fig 2A. After 2 hrs of infection the explants were washed and incubated for 48 hrs in liquid media. Culture supernatants were collected at 24 hrs and 48 hrs for viral replication assessment by qRT-PCR. Each experiment was set up in triplicate where possible depending on the size of the tissue received. The error bars represent the standard deviation between experiments. The dotted line represents the limit of detection (50 copies / reaction). NV = neat virus, Control = uninfected explants.
(TIF)

**S12 Fig.** *Ex vivo* **challenge of oral and nasal tissue from study participants 17 and 18 with SARS-CoV-2.** Two or four oral or nasal tissue explants from participants 17 and 18 were infected with SARS-CoV-2, strain IC19 using the modified protocol as described in Fig 2A. After 2 hrs of infection the explants were washed and incubated for 48 hrs in liquid media. Culture supernatants were collected at 24 hrs and 48 hrs for viral replication assessment by qRT-PCR. Each experiment was set up in triplicate where possible depending on the size of the tissue received. The error bars represent the standard deviation between experiments. The dotted line represents the limit of detection (50 copies / reaction). NV = neat virus, Control = uninfected explants.
(TIF)

**S1 Table. Primers and probe sequences (with modifications) for variant calling and cDNA synthesis of Wuhan, Beta and Delta lineages of SARS-CoV-2.**
(DOCX)

**S1 File.**
(XLSX)

**S2 File.**
(XLSX)

**S3 File.**
(XLSX)

## Acknowledgments

We thank Khaleel Ismail, Ziyaad Waja and the staff at the Donald Gordon Hospital for recruiting participants, collecting, and transporting the tissue samples to the laboratory. We are grateful to the participants for providing their tissue samples. We thank Wendy Barclay (Imperial College) for providing the SARS-CoV-2 viral stocks, and Prof. Wolfgang Preiser and Dr Tasnim Suliman (University of Stellenbosch) for sharing stocks of the Vero E6 cells.

## Author Contributions

**Conceptualization:** Carolina Herrera, Francesca Chiodi, Neil Martinson, Julie Fox, Bavesh Davandra Kana.

**Data curation:** Bhavna Gowan Gordhan, Azure-Dee Pillay, Thabiso Seiphetlo, Natasha Singh, Kennedy Otwombe.

**Formal analysis:** Bhavna Gowan Gordhan, Bavesh Davandra Kana.

**Funding acquisition:** Neil Martinson, Julie Fox, Bavesh Davandra Kana.

**Investigation:** Bhavna Gowan Gordhan, Azure-Dee Pillay, Thabiso Seiphetlo, Natasha Singh.

**Methodology:** Bhavna Gowan Gordhan, Carolina Herrera, Christopher Shawn Ealand, Edith Machowski, Gabriella Scarlatti, Bavesh Davandra Kana.

**Project administration:** Bhavna Gowan Gordhan.

**Resources:** Carolina Herrera, Ntombexolo Seatholo, Limakatso Lebina, Rebecca Frise, Neil Martinson, Bavesh Davandra Kana.

**Supervision:** Bhavna Gowan Gordhan, Bavesh Davandra Kana.

**Writing – original draft:** Bhavna Gowan Gordhan, Bavesh Davandra Kana.

**Writing – review & editing:** Bhavna Gowan Gordhan, Carolina Herrera, Azure-Dee Pillay, Thabiso Seiphetlo, Christopher Shawn Ealand, Edith Machowski, Natasha Singh, Kennedy Otwombe, Limakatso Lebina, Francesca Chiodi, Neil Martinson, Julie Fox, Bavesh Davandra Kana.

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
