## [Decision Letter · Decision Letter 0]

7 Jul 2023

PONE-D-23-15626Evaluation of a human mucosal tissue explant model for SARS-CoV-2 replicationPLOS ONE

Dear Dr. Kana,

Thank you for submitting your manuscript to PLOS ONE. After careful consideration, we feel that it has merit but does not fully meet PLOS ONE’s publication criteria as it currently stands. Therefore, we invite you to submit a revised version of the manuscript that addresses the points raised during the review process.

We look forward to receiving your revised manuscript.

Kind regards,

Ralph A. Tripp

Academic Editor

PLOS ONE

Journal Requirements:

3. Please expand the acronym “EDCTP” (as indicated in your financial disclosure) so that it states the name of your funders in full.

5. We note that Figures 1 and 2 in your submission contain copyrighted images. All PLOS content is published under the Creative Commons Attribution License (CC BY 4.0), which means that the manuscript, images, and Supporting Information files will be freely available online, and any third party is permitted to access, download, copy, distribute, and use these materials in any way, even commercially, with proper attribution. For more information, see our copyright guidelines: http://journals.plos.org/plosone/s/licenses-and-copyright.

a. You may seek permission from the original copyright holder of Figures 1 and 2 to publish the content specifically under the CC BY 4.0 license. 

Additional Editor Comments:

please address the reviewer comments.

Reviewers' comments:

Reviewer's Responses to Questions

**Comments to the Author**

1. Is the manuscript technically sound, and do the data support the conclusions?

Reviewer #1: Partly

Reviewer #2: Yes

2. Has the statistical analysis been performed appropriately and rigorously? 

Reviewer #1: No

Reviewer #2: Yes

3. Have the authors made all data underlying the findings in their manuscript fully available?

Reviewer #1: No

Reviewer #2: Yes

4. Is the manuscript presented in an intelligible fashion and written in standard English?

Reviewer #1: Yes

Reviewer #2: Yes

5. Review Comments to the Author

Reviewer #1: The study addresses the issue to disease modeling in Covid infection with the ultimate goal to develop a reliable, reproducible and clinically relevant model of virus entry, replication and biological effects. Such a model could be useful in studying viral characteristics such as entry and life-cycle, pathogenic features but most importantly evaluating the efficacy of novel drugs. Disease modeling using human samples is of particular importance in the absence of suitable animal models for SARS-CoV-2 infection. Among well accepted approaches in the literature are explant cultures of target tissues for the virus, primarily lung tissue. Since the virus can enter the body via and be detected in significant amounts in the nasal and oral cavities, the investigators have used for the explant experiments residual oral and nasal tissue obtained from 18 individuals undergoing elective maxillofacial surgery. It is not clear the type of epithelium present in those tissues and perhaps a more detailed description, some representative H&E stained pictures or adding a patient table might have contributed to a better understanding of the experimental approach. The tissue was transferred within one hour of its harvesting to the lab which is an advantage for the overall tissue survival. It is not mentioned whether the tissues were frozen for the experiment or used as fresh material. The investigators quite appropriately used titration experiments for the viral input. Overall they should be commented for presenting in detail and in a comprehensive way (supported by diagrammatic study overview for each experiment) their different approaches to establish viral replication in an explant system and providing adequate rational for pursuing several protocol modifications. Appropriate controls have been included. Unfortunately, the team was not able to demonstrate significant viral replication in their system and in our opinion, publication of negative data should be encouraged. However, as it is currently structured the manuscript appears rather descriptive, particularly data presented in Fig 3. For example, one of the strengths of the explant approach is that is allows studying patient heterogeneity. For example, it would be of interest in Fig 1 the results to be presented for each patient individually (a panel of 10 datasets) and also clarify whether the tissue was of oral or nasal origin. Same is true for Fig 2. Furthermore, data analysis should be performed in relation to information about tissue histology and additional patient characteristics that might potentially impact tissue survival and suitability for explant infection. It has been previously reported that the survival of epithelial cells from the gastrointestinal tract is substantially declining after 24 hours in culture. Which exact type of cells are the investigators aiming at infecting with the virus and under which kinetics assumptions? In Fig 3 a diagram of the experimental protocol should be added, as in Fig 1-2, in order to facilitate the interpretation of the results. It would be helpful if the authors could add a working hypothesis for the experiment described in Fig 3. Are the neutralizing antibodies originating from residual blood in the tissue or are produced de novo from infiltrating B-cells? In its current form, data presented in Fig 3 is rather “anecdotal” since only two patients have been profiled. Again, in order to gain any meaningful conclusions, all available post-wash samples should be tested using the same methodology and data from all available patients should be presented in Fig 3. In Fig 3, what is the NC (non-culture supernatant) control? Is it culture media with serum? Data presented in Fig 3 is puzzling from a technical point of view as well. For instance, in vaccinated participant 1, why in NC control incubation at 65oC enhanced viral replication and at the same time reduced viral replication in the CS? The plaque assay should be run at least in triplicate for each patient due to method variability. In vaccinated participant 2, a similar concern was raided: There is apparent viral replication enhancement under all conditions, irrespectively of explant secreted factors. Those concerns should be adequately addressed by the investigators or if not possible due to technical reasons, the neutralization assessment experiments should be taken out of this manuscript till the methodology is working properly based on positive and negative controls.

Discussion needs to be extended and specifically address how the methodology used in this study (Fig1 &2) compares to previous publications using nasal explants (REF 11,13).

Reviewer #2: The authors have conducted a clinically relevant study with translational potential. The study design is appropriate for the study question and the scientific weakness has been properly identified and mentioned. The data analysis and interpretation and discussion of the outcome is scientifically sound. The authors have not mentioned what precautions were taken in the lab or during the experiments to prevent transmission of the infection to personnel. The authors have also not mentioned the source of the virus. Both these aspects are relevant in case the study leads to further studies of a similar nature looking at any other aspects of the topic/subject matter.

6. PLOS authors have the option to publish the peer review history of their article (what does this mean?). If published, this will include your full peer review and any attached files.

Reviewer #1: No

Reviewer #2: No

---

## [Author Response · Author response to Decision Letter 0]

21 Aug 2023

The reviewer and editors comments have been addressed and has been uploaded as a Word file called "Response to reviewers"

---

## [Editor Report · Decision Letter 1]

23 Aug 2023

Evaluation of a human mucosal tissue explant model for SARS-CoV-2 replication

PONE-D-23-15626R1

Dear Dr. Kana,

We’re pleased to inform you that your manuscript has been judged scientifically suitable for publication and will be formally accepted for publication once it meets all outstanding technical requirements.

Kind regards,

Ralph A. Tripp

Academic Editor

PLOS ONE

Additional Editor Comments (optional):

Despite the negative results for some studies, the authors have satisfactorly addressed the reviewers concerns and the manuscript is now acceptable.
---

## [Editor Report · Acceptance letter]

19 Sep 2023

PONE-D-23-15626R1 

Evaluation of a human mucosal tissue explant model for SARS-CoV-2 replication 

Dear Dr. Kana:

I'm pleased to inform you that your manuscript has been deemed suitable for publication in PLOS ONE. Congratulations! Your manuscript is now with our production department. 

Kind regards, 

on behalf of

Dr. Ralph A. Tripp 

Academic Editor

PLOS ONE